# Prevention of Encrustation on Ureteral Stents: Which Surface Parameters Provide Guidance for the Development of Novel Stent Materials?

**DOI:** 10.3390/polym12030558

**Published:** 2020-03-03

**Authors:** Henrike Rebl, Jürgen Renner, Wolfgang Kram, Armin Springer, Nele Fritsch, Harald Hansmann, Oliver W. Hakenberg, J. Barbara Nebe

**Affiliations:** 1Dept. of Cell Biology, Rostock University Medical Center, Schillingallee 69, 18057 Rostock, Germanybarbara.nebe@med.uni-rostock.de (J.B.N.); 2Institute for Polymer Technologies e.V. (IPT), Alter Holzhafen 19, 23966 Wismar, Germany; renner@ipt-wismar.de (J.R.); h.hansmann@ipt-wismar.de (H.H.); 3Dept. of Urology, Rostock University Medical Center, Ernst-Heydemann-Straße 6, 18057 Rostock, Germany; wolfgang.kram@med.uni-rostock.de (W.K.); oliver.hakenberg@med.uni-rostock.de (O.W.H.); 4Electron Microscopy Center, Rostock University Medical Center, Strempelstraße 14, 18057 Rostock, Germany; armin.springer@med.uni-rostock.de

**Keywords:** ureteral stent, encrustation, polymer, urothelial cells, zeta potential, wettability

## Abstract

Encrustations of ureteral stents are one of the biggest problems with urological implants. Crystalline biofilms can occur alone or in combination with bacterial biofilms. To identify which surface parameters provide guidance for the development of novel stent materials, we used an in vitro encrustation system. Synthetic urine with increasing pH to simulate an infection situation was pumped over the polymer samples with adjusted flow rates at 37 °C to mimic the native body urine flow. Chemical surface features (contact angle, surface charge), as well as encrustations were characterized. The encrustations on the materials were analyzed quantitatively (dry mass) and qualitatively using scanning electron microscopy (SEM), energy dispersive X-ray spectroscopy (EDX), and Fourier transform infrared spectroscopy (FTIR). The aim of this comparative study was to identify crucial surface parameters that might predict the quantity and type of mineral deposits in vitro and provide guidance for the development and screening of new polymer-based biomaterials for ureteral stent design. For the first time, we could identify that, within the range of our polymers, those materials with a slight hydrophilicity and a strong negative zeta potential (around −60 mV) were most favorable for use as ureteral stent materials, as the deposition of crystalline biofilms was minimized.

## 1. Introduction

Ureteral stents are used for temporary, as well as long-term stenting of the ureter. Despite the risks associated with the use of stents, urinary stents are one of the most frequently used human implants. Complications of stents, like biofilm formation or encrustation, promote urinary tract infections and cause disorders in wound healing [1]. Stent-associated urinary tract infections are among the most common (25% to 40%) nosocomial infections [2,3]. Additional dangers are the frequently occurring encrustations of the stents that are triggered by the precipitation of salt crystals [4].

The development of urinary stones in patients has multiple causes, whereby genetic, dietary, climatic, and socioeconomic factors play a role [5]. In the absence of an infection, the gender-spanning dominant urinary stone type is calcium-containing stones [6], mainly calcium oxalate monohydrate and, to a lesser extent, calcium oxalate dihydrate. Such calcium-containing precipitations are even found on the stents of patients with a normal urine calcium level and no prior kidney stones [7].

However, in the case of an infection, especially with urease-producing bacteria, urinary stones predominantly consist of struvite (magnesium phosphate) and carbonate apatite and are often called “infection stones” [8]. In this case, the pH is elevated due to the decomposition of urea by the enzyme urease, which can be produced by bacteria like *Proteus* spp., *Klebsiella* spp., or some strains of *Escherichia coli* [9]. The change of the pH alters the solubility of magnesium and calcium salts, leading to the precipitation of salt crystals. This phenomenon is used by different authors in established encrustation models to simulate in vitro stone formation [8,10,11].

The pH increases, and thus, the formation of more encrustations, which in turn promote the adhesion of bacteria, leads to a vicious cycle with massive encrustations and biofilm formation occurring at the same time in the patient [5]. Aggregation of the crystalline deposits occurs in the urine, on the stents’ surface, and in the developing bacterial biofilm. This leads to crystalline bacterial biofilms, with an accumulation of numerous microorganisms that are embedded into a matrix of metabolites and crystallized urinary components. The risk of stent-associated infection increases with catheterization time, and over 90% of patients undergoing long-term catheterization, especially with indwelling catheters, will develop bacteriuria [12].

The choice of new materials and coating processes is significantly restricted by the necessary basic requirements for medical implants (e.g., bio- and hemo-compatible, chemically inert, low friction). Various catheter concepts have been tested in many divergent in vitro and in vivo test series [13]. The comparability of the experimental results is therefore only possible to a limited extent, and the discussions are sometimes very controversial [14]. For the direct comparison of novel polymer materials, the systematic performance of encrustation tests with different urological stents in one in vitro encrustation model is absolutely mandatory. Therefore we aim at determining surface parameters that might predict the outcome of a material with respect to the encrustation behavior. 

We tested substrates of different polymer types (thermoplastic urethanes, styrene-butadiene copolymers, ethylene-vinyl acetate copolymers) and determined the chemical characteristics (contact angle, surface charge), as well as encrustations. We developed an in vitro encrustation system, where chemically defined synthetic urine was dynamically pumped over the samples at 37 °C to mimic the native body urine flow. The encrustations were analyzed quantitatively and qualitatively. 

The aim of this comparative study was to identify, for the first time, crucial surface parameters that might be correlated with the quantity and type of mineral deposits in vitro and thus provide guidance for the development of new biomaterials that can be used in ureteral stent design. Finally, the goal is to develop new polymer materials for urinary stents that should minimize encrustation, prolong the indwelling time, and thus, reducing the burdens on the patient. 

## 2. Materials and Methods

### 2.1. Polymer Materials

We tested substrates of different polymer types (see Table 1). The polymers were purchased as granules. Elastollan was purchased from BASF SE (Ludwigshafen (Rhein), Germany), Styroflex from INEOS Styrolution Group GmbH (Fankfurt am Main, Germany), and all other polymers from Lubrizol (Wickliffe, OH, USA).

A heating press from VOGT Maschinenbau GmbH (Berlin, Germany) was used to produce flat samples. The temperature control to the target value for the upper and lower plates was carried out separately. The pressing process of the pre-dried granulate was carried out with the aid of intermediate layers of Polytetrafluorethylene-coated glass fabric Type 7069 from TACONIC (Rensselaer, NY, USA). Spacers between the glass fabrics were used to achieve a defined layer thickness of the flat samples. The processing temperature for the respective polymer is shown in Table 1.

After the press plates had reached the spacers, a residence time of 30 s was observed to ensure complete melting of the polymer. The intermediate layers were then removed and cooled in a water bath. After cooling, the sample could be removed easily from the intermediate layers.

### 2.2. Surface Characteristics 

The pure polymer samples were cleaned with distilled water and air-dried, and the following surface parameters were determined.

#### 2.2.1. Contact Angle

The contact angle between a drop of liquid and a flat surface is a measure of the wettability of this surface. Synthetic urine with pH 6.5 (Section 2.3; but without the addition of urease to prevent crystal formation) was used as the test liquid. Contactless, optical measurement of the contact angle was carried out with a Drop Shape Analyzer (DSA100, Krüss, Hamburg, Germany). The ambient temperature of the sample and the test liquids was 20 °C. Within 45 s, after droplets with a volume of 5 µL had settled, the contact angle was measured at an interval of 0.5 s, averaging the left and right sides of the recorded angle. This procedure ensured a proper energy equilibrium between the sample and test liquid, whereby evaporation did not affect the measurement. The measured values were interpolated at the curves in the area of the quasi static equilibrium. 

#### 2.2.2. Surface Charge

A descriptive parameter of surface chemistry is the zeta potential that develops at the interface between a solid and a surrounding liquid. It describes a specific surface charge that is generated in the presence of an aqueous solution. Experimental access to the zeta potential of macroscopic solid surfaces is achieved by measuring the flow potential and the flow current. Under defined pressure conditions, the solid is inundated with an aqueous solution. 

Zeta-potential measurements were conducted on polymer samples (2 cm × 1 cm) using the SurPASS^TM^ system (Anton Paar, Ostfildern, Germany) to determine the surface charge. The measurements were performed in a 0.001 mol/L KCl solution ranging from pH 5.5 to 8.0 with a gap height of 100 µm between two samples. The streaming current was determined depending on the pressure (max. 400 mbar). Finally, the zeta potential was calculated according to the method of Helmholtz–Smoluchowski. Measurements of each polymer were performed in quadruplicate on three independent sample pairs.

### 2.3. Encrustation

The setup of the in vitro encrustation system is shown in Figure 1. Chemically defined synthetic urine was prepared according to Griffith et al. [8]. In brief, Solutions A and B were prepared with distinct amounts of different salt components. Solution A was supplemented with 0.3 U/mL urease (Carl Roth GmbH, Karlsruhe, Germany) to induce the crystallization process. Three hundred thirty three milligrams per liter of bovine serum albumin (Serva Electrophoresis GmbH, Heidelberg, Germany) were added to Solution B as a protein component. The temperature in the incubation system was set to 37 °C. The flow rate (Ismatec IPC pump system, Cole Parmer GmbH, Wertheim, Germany) was adjusted to 150 µL/min, with stop-go cycles of 50 s/10 s each to obtain measurable encrustations after 5 days in the encrustation system. The encrustations were analyzed quantitatively by measuring the weight of the samples before and after the experiment. The samples were allowed to dry for 3 d at 37 °C to ensure that all residual liquid was evaporated. 

#### 2.3.1. Scanning Electron Microscopy

Qualitative analyses were performed using field emission scanning electron microscopy (FE-SEM, Merlin^®^ VP compact, Zeiss, Oberkochen, Germany). The samples were mounted on an SEM carrier with adhesive conductive carbon tape (PLANO, Wetzlar, Germany). Surface conduction was improved by sputtering with a gold layer (approximately 1520 nm thickness) using a Bal-Tec SCD004 sputter coater (Balzers Union Ltd., Balzers, Liechtenstein). The SEM was operated at 5 kV.

#### 2.3.2. Energy Dispersive X-ray Spectroscopy 

Samples were analyzed by a field emission scanning electron microscope (FE-SEM, Merlin^®^ VP Compact, Zeiss, Oberkochen, Germany) equipped with an energy dispersive X-ray (EDX) detector (XFlash 6/30, Bruker, Berlin, Germany). The samples were coated with carbon under vacuum (EM SCD 500, Leica, Bensheim, Germany). Representative areas of the samples were analyzed and mapped for elemental distribution on the basis of the EDX-spectra data by QUANTAX ESPRIT microanalysis software (Version 2.0, Bruker, Billerica, MA, USA). EDX mappings were taken from the selected regions operated with 10 kV accelerating voltage and 500 times magnification.

#### 2.3.3. Fourier Transform Infrared Spectroscopy 

Encrustations deposited on the polymer samples were dried before the analysis and were ground in an agate mortar. The analysis was done by the Urinary Stone Analyzer ALPHA FTIR spectrometer system (Bruker, Billerica, MA, USA) with the ATR (attenuated total reflection) measurement interface. Spectra were recorded between 4000 and 400 cm^−1^ (resolution 2 cm^−1^) for 120 scans and compared to the OPUS™ reference library, Version 7.5 (Bruker, Billerica, MA, USA). Semi-quantitative results were recorded in percentages. The qualitative analysis was based on the comparison of the “spectral fingerprint” of a sample with the spectrum of the reference material (synthetic urine).

### 2.4. Cell Culture

#### 2.4.1. Cells

The biocompatibility of the polymer samples was studied using the human non-tumorigenic urothelial HUC-1 cells (ATCC, CRL-9520, LGC Standards GmbH, Wesel, Germany). Cells were cultivated in Dulbecco’s Modified Eagle Medium (DMEM, Thermo Fisher Scientific, #10569, Waltham, MA, USA) with 10% fetal calf serum (FCS, PAN Biotech GmbH, Aidenbach, Germany, Lot No. P281305) and 1% antibiotic-antimycotic (Thermo Fisher Scientific, #15240062, Waltham, MA, USA) at 37 °C in a humidified atmosphere with 5% CO_2_.

#### 2.4.2. Biocompatibility Assay

The MTS assay (CellTiter 96^®^ Aqueous One Solution Cell Proliferation Assay, Promega, Madison, WI, USA) was performed to assess the biocompatibility of the polymers. The principle of the test is an enzymatic cleavage of the methyltetrazolium salt (3-(4,5-dimethylthiazol-2-yl)-5-(3-carboxymethoxyphenyl)-2-(4-sulfophenyl)-2H-tetrazolium, inner salt (MTS)) by metabolically active cells into a formazan product. The amount of formazan product is directly proportional to the number of living cells. 

Then, 1.5 × 10^5^ cells were seeded into 24-well cell culture plates (TCPS, Greiner, Kremsmünster, Austria), and the polymer samples (6 mm diameter) were added via ThinCert inserts (8 µm pore size, Greiner, Kremsmünster, Austria) at a volume of 600 µL. After 24 h, 100 μL of the MTS solution were added to each well and incubated for 2 h. Afterwards, 100 µL of the supernatant were transferred into a 96 well plate in triplicate, and the spectrophotometric absorption was analyzed by an ELISA reader (Anthos 2010, Anthos Labtec Instruments, Wals, Austria) at 490 nm.

### 2.5. Statistics

Statistical analyses were performed with the software GraphPad PRISM 7 (GraphPad Software, San Diego, CA, USA) using the ANOVA post hoc Bonferroni or Kruskal–Wallis post hoc uncorrected Dunn’s test. Data were presented as the mean ± standard deviation (SD); a probability value of *p* < 0.05 was considered significant.

## 3. Results and Discussion

The metabolic make-up of urine is the determining factor in encrustation behavior. In an infection situation, the pH is elevated, and this alters the solubility of magnesium and calcium salts, leading to more magnesium phosphate encrustations [9]. 

Besides this, there are other factors that account for a change of stents in the daily clinical routine. Laube et al. investigated 59 stents that were removed from the patients because of clinically observed “stent obstruction” [15]. This phenomenon is often described by the fact that the urine flow is compromised due to large depositions in the lumen of the stent. However, the authors found that 75% of the investigated stents showed no sign of inner depositions. In these cases, the reason for the hampered urine flow is not clear. It can be speculated that the problems might be related to the stent’s design (collapsed lumen, inadequate diameter) or material composition (swelling, inelastically compressed). We wanted to address the relationship between the physical properties of a polymer surface and its ability to resist encrustation. If we can identify desirable physical surface characteristics, it might be possible to manipulate polymer compositions and develop materials that resist encrustation even after prolonged exposure to urine. 

First, we tested all of the polymers for their biocompatibility, as this is one main prerequisite for novel materials. The biocompatibility of the polymers was tested with non-tumorigenic human urothelial cells HUC-1 (Figure 2). None of the polymers elicited toxic effects, and thus, all tested polymers might be used for the design of urinary stents.

### 3.1. Use of Synthetic Urine to Ensure Reproducibility in Comparative Studies

The formation of urinary stones in vivo is not well understood and depends on a multitude of factors like pH, protein excretion, the presence of stent/foreign material, bacterial infection, or dietary habits [5]. Therefore, in vitro encrustation models have been developed [8,10,16] to enable systematic studies that might help to gain a deeper understanding of the problem. We built our in vitro encrustation system based on this knowledge and used a synthetic urine introduced by Griffith [8]. In this approach, two different, undersaturated solutions were pumped into a reaction chamber in which the polymer samples were placed (Figure 1). The initial pH of the synthetic urine was 5.6. Over time, the urease decomposed the urea into carbon dioxide and ammonia. This led to a shift in the pH to alkaline conditions, causing the precipitation of salt crystals [8]. The rate of pH increase caused by the urease reaction, as well as the pH value reached in the equilibrium state (pH 8.5) realistically simulated an in vivo infection situation (Appendix A). The use of defined amounts of urease, rather than adding urease-producing bacteria, allowed for more reproducible experimental conditions.

Kleinen et al. [10] found that the quality and quantity of encrustations depended furthermore on the protein amount; they came to this conclusion using different concentrations of bovine serum albumin in their experiments. Serum albumin is the main component of the organic matrix in calcium phosphate crystals precipitated from urine collected from stone forming patients [17]. Based on these findings, we decided to use albumin at a concentration of 333 mg/L in our in vitro system. 

### 3.2. Surface Characteristics can be Correlated to Encrustation Mass

Contact angle: The contact angle is a measure to describe the hydrophilicity/hydrophobicity of a material. In general, surfaces with a contact angle of <90° are referred to as hydrophilic, whereas materials with a contact angle >90° are referred to as hydrophobic [18]. To mimic the in vivo situation as far as possible, we performed contact angle measurements with the synthetic urine described above. The measurements revealed the values shown in Table 2. Tecoflex was determined to be the most hydrophobic material with 111°, while Styroflex and Elastollan were slightly hydrophilic with 85° (see also Appendix A).

Zeta potential: ζ potential measurements (Table 2) detected negative charges for all examined materials. Table 2 shows the zeta potential at pH 6.5, representing the average urine pH of healthy humans. Tecophilic had the smallest negative charge with −29.6 mV, while Styroflex and Elastollan were strongly negatively charged with −66.6 mV and −56.5 mV, respectively.

The slope of the curves in Figure 3 reveals slightly more negative values at higher pH, as would occur in an infection situation. Even though the absolute values differed between the investigated polymers, the slope of the curves seemed similar for all materials. 

Based on the values at pH 6.5, we could divide our polymers into two groups: Group 1: hydrophobic (above 90°) and slightly negatively charged (Carbothane, Pelletane, Tecoflex, Tecophilic, Tecothane)Group 2: moderately hydrophilic (around 85°) and a strong negative surface charge (Elastollan, Styroflex, and Greenflex).

The different biomaterials were incubated in the in vitro test system for five days at 37 °C. After that, the samples were dried, and the net weight increase was quantified (Figure 4). The lowest encrustation mass could be measured on Styroflex, Elastollan, and Greenflex (2.1%, 2.4%, and 2.8%, respectively). In contrast, Pelletane and Tecophilic were highly encrusted (4.0% and 6.0%). 

Our comparative analyses showed that the low quantity of encrustation could be correlated with a strong negative surface charge and good hydrophilicity of the polymer (Group 2). This means that the least encrusted material (Styroflex) was also the one material with the strongest negative surface charge. Moreover, it was the material with the second lowest contact angle amongst the investigated polymers.

The attraction or repulsion of the salt crystals depended on the electrostatic interaction between the surface and the crystalline deposits. Equal charges repelled each other, while oppositely charged surfaces attracted each other. The main components of infectious urinary stones were struvite (magnesium phosphate) and carbonate apatite. Carbonate apatite crystals possessed a negative surface charge (−16 mV at pH 8.5), as shown by Prywer et al. [19]. Furthermore, Le Corre found that struvite crystals also exhibited a negative zeta-potential of −17.5 ± 1.1 mV at a pH of 8.5. The scope of their work was to identify binding partners for struvite crystals during the crystallization process. They identified the positively charged polyDADMAC as a good polymer to bind struvite particles. The opposite charges of polyDADMAC and struvite ensured a strong binding of the two molecules [20]. 

Furthermore, there were indications that coating of ureteral stents with glucosaminoglycans [21] or heparin [22] (both negatively charged) resulted in decreased biofilm formation and encrustations. 

As a consequence, a surface that has a strong negative surface charge would repel these salt crystals and would thus be more resistant to encrustation than a positively charged surface. With regard to our investigated polymers, struvite and apatite crystals were probably more strongly repelled by the strongly negatively charged surfaces (Group 2, Elastollan, Styroflex, and Greenflex). 

This mechanism might explain why those investigated polymers with strong negatively charged surfaces displayed less depositions in our encrustation experiments.

Commercially available stents are mostly made of silicone or polyurethanes [23]. Polyurethanes are a broad class of polymers that may differ in their chemical surface characteristics, like wettability or surface energy. Unfortunately, the manufacturers do not state exactly which polyurethane is used for their specific stent. Laube et al. investigated one commercially available double-J-stent (without mentioning the manufacturer’s name) and determined a water contact angle of 103°+/-3° for this polyurethane stent [11]. After deposition of a hydrophilic (78°) amorphous carbon coating, the encrustation rate could be reduced drastically. The authors stated that this hydrophilic coating facilitated the fast lift-off of nanometer-sized gas bubbles from the surface, taking precipitated salt crystals with them, thus resulting in a “self-cleaning” mechanism that slowed down the encrustation process. To come closer to a real in vivo situation, Bonner et al. determined the contact angle of a polyurethane sheet at 26° after 24 h immersion in human urine, unfortunately without mentioning which type of polyurethane was investigated [24]. Our approach was to use synthetic urine for the determination of the contact angle to come closer to the in vivo situation while maintaining the reproducibility given by the selective composition of synthetic urine. 

### 3.3. Characterization of Encrustations: Deposits Resemble In Vivo Infection Situation

Scanning electron microscopy and EDX: Next, we investigated the morphology and composition of the encrustations on the different polymers. The polymer samples were incubated in synthetic urine for five days. We wanted to simulate an infected in vivo condition, where elevated pH was generated by the enzymatic cleavage of urea. In vivo, this leads to the precipitation of carbonate apatite and struvite stones [8]. We wanted to find out whether different polymer surface properties might evoke differences in mineral depositions directly on the material surface. 

On the material’s surface, we found carbonate apatite occurring as microcrystalline aggregations and struvite appearing as rather large, coffin-shaped crystals [1]. However, the distribution of these depositions differed drastically on the polymers investigated (Figure 5). 

It was found that the three polymers with the lowest encrustation mass (Styroflex, Elastollan, and Greenflex) showed very smooth depositions resembling a cauliflower-like morphology. Large areas of the samples were covered almost exclusively with this smooth film of crystallites. Only very few coffin shaped large crystals were found on these surfaces. We used EDX to determine the elemental composition of these depositions (Figure 6). The smooth layer was shown to be composed mainly of Ca, P, C, and O, presumably being calcium carbonate apatite (Ca_5_[OH|(PO_4_)_3_]) crystallites. The few large crystals were composed of Mg, P, C, and O, which could be assigned to struvite (magnesium phosphate; NH_4_MgPO_4_). 

The other polymers (Carbothane, Pelletane, Tecoflex, Tecophilic, Tecothane) were highly encrusted (Appendix A). We found a mix of large struvite crystals on top of or in between the hydroxyapatite layer, which rendered the resulting surface very rough. Some of the crystals reached sizes of up to 100–150 µm in length with sharp-edged tips. Cox et al. analyzed stents removed from patients with infections and described the encrustations as large struvite crystals embedded in a crust of hydroxyapatite crystallites [25]. This morphology can also be seen on the highly encrusted polymers investigated.

We believe that the morphology of the crystals was relevant for the evaluation of the in vivo situation. Sharp-edged crystals might cause irritations of the urothelial barrier and should be avoided. More importantly, large agglomerates of these crystals can create a niche for bacterial growth by slowing down the urine flow. Adherent bacteria in these shadow zones of the crystals are thereby protected from being rinsed away by the urine stream. 

Fourier transform infrared spectroscopy (FTIR): FTIR can be used to obtain an infrared spectrum of absorption or emission of a solid sample. The IR spectrum of a sample is the fingerprint of the molecular species making up the sample. The quantification of the relative amount of each constituent without using any solvent makes FTIR the most appropriate approach for stone analyses [26]. We analyzed the composition of the mineral deposits of the synthetic urine without polymer and the precipitates on the surface of the polymer samples using FTIR spectrometry (Figure 7). In this study, we used synthetic urine with a defined composition that resulted in mineral deposits similar to those produced on stents under in vivo infection situations. The absorption curves of crystals from the different polymer materials were recorded and compared to the absorption curves of the OPUS™ reference library (Bruker). The curve with the best fit was chosen for the determination of the percentual composition. In the precipitates of the synthetic urine, we detected a mixture of 45% carbonate apatite, 35% struvite, and 20% protein. As described before, carbonate apatite and struvite (magnesium phosphate) were found in urinary “infection stones”. 

These crystalline deposits were also described by others, however without determining the exact amount of each crystal species [8,27]. 

Next, we analyzed the precipitates from the encrusted polymer samples. In this study, we used the ALPHA FTIR spectrometer by Bruker that is additionally equipped with the OPUS^TM^ library. In this library, a comparison with a collection of spectra (approximately 5000 reference spectra of realistic stones) allowed a statement about the qualitative composition of the urinary stone samples. Because of this, we did not concentrate on the discrete formation of the absorption bands for our analyses, but rather focused on the correspondence of the measured spectra with the filed reference spectra. This correspondence was indicated by the “hit quality”, where 1000 meant a perfect match. Our spectra showed a hit quality of ~700–800. This means that even though we used synthetic urine, we produced crystals that came quite close to the in vivo situation. Using this method, we could now compare the stone composition between our different samples (for spectra of all samples, see Appendix A).

On the sparsely encrusted sample of Elastollan, the quantification determined the best fit with a mixture of 60% whitlockite, 20% carbonate apatite, and 20% struvite. Whitlockite (tricalcium phosphate Ca_3_(PO_4_)_2_) is a crystal of calcium orthophosphate in which magnesium is partly substituted for calcium. These crystals commonly form in biologic systems because of the high concentrations of proteolipids and divalent cations in biologic fluids [28]. Overall, we had a major phase of calcium phosphate-containing stones (carbonate apatite and whitlockite) on the material surface, compared to the precipitates from the synthetic urine without polymer.

In contrast, the spectrum recorded on the highly encrusted Tecophilic polymer had the highest match with the absorption spectrum of a composition of 50% carbonate apatite, 40% struvite, and 10% whewellite (hydrated calcium oxalate). Here, we found a much higher rate of struvite (magnesium phosphate)-containing stones in comparison with the precipitates from the synthetic urine without polymer.

Altogether, the quantification revealed that, as indicated by the SEM analyses, the amount of struvite crystals was higher on the severely encrusted surfaces (Group 1: Tecophilic, Carbothane, Pelletane), whereas on the minimally encrusted polymers (Group 2), we found mainly calcium phosphates. As we could also correlate the surface parameters’ wettability and surface charge to this classification (minimally encrusted materials with intermediate wettability 85° and with a strong negative surface charge around −60 mV), we therefore assumed that these surface characteristics played a major role in the adhesion and formation of crystalline deposits.

In experiments with human urine collected from patients, Griffith et al. obtained qualitatively similar results to those obtained with the synthetic urine used in this study [8]. In the clinical routine, approximately 10% of all urinary stones were made up of struvite [29] and originated from infection by bacteria that possessed the enzyme urease [15]. Crystallographic analyses showed that often, human “struvite” stones were not monocrystalline, but a mixture of struvite (MgNH_4_PO_4*_6H_2_O) and carbonate apatite (Ca_10_[PO_4_]_6*_CO_3_) with trace amounts of calcium oxalate [30]; this made exact classification very complicated.

In this study, we used a screening model to mimic those encrustations that occurred in an infection situation; thus, we aimed at producing encrustations of struvite and carbonate apatite. We were aware that we were using a model test system that was lacking some components of real human urine (like Tamm–Horsfall glycoproteins, parathormone, uropontin, nephrocalcin) and other components of an infection (living bacteria or components of the human immune system). However, the use of the selected key parameters and components that cause encrustations enabled us to perform highly reproducible experiments with various biomaterials. The goal should be to reduce animal experiments at a time when the number of available polymers and admixtures is increasing rapidly.

The investigation of new materials for urinary stents is a key step to optimize the clinical outcome. Current stents are prone to encrustations and biofilm formation. 

The use of newly designed polymer stents might offer new strategies for stent development. The goal in biomaterials research must be to find new materials that decrease crystalline biofilm formation and improve patient comfort by reducing complications and increasing the indwelling times, finally leading to lower morbidity. The inclusion of such an encrustation model in the standards for testing materials should lead to better biomaterials being developed [16]. 

Our systematic analyses showed, for the first time, that the measurement of wettability and surface charge could give an indication of a material’s tendency to encrust. Especially surface charge measurements could assist to assess the outcome of ureteral stents, and its importance has been widely underestimated. We recommend materials with an average hydrophilicity of around 85° and a strong negative surface charge for use in urology. A large number of new materials can thus be tested quickly and effectively. Manufacturers of urological implants can use these data to select biomaterials that resist encrustation processes while the stent is in the patient and to produce implants that help patients. The traditional structure of a urological stent as a solid smooth tube might be questioned as new materials will allow for innovative stent designs. The combination of these minimally encrusting materials with antimicrobial agents may offer new opportunities.

## 4. Conclusions

Here, we presented a systematic screening approach that should describe which material surface parameters indicate low encrustation characteristics. We developed an in vitro encrustation system in which we were able to provoke encrustation on polymer samples within five days.

We showed that, within our tested materials, those materials with a slight hydrophilicity (around 85°) and a strong negative surface charge (around −60 mV) were most favorable for use as stent material, as the deposition of crystalline biofilms was minimized. 

We showed for the first time that the morphology and quantity of the encrustations (highest Tecophilic > Greenflex > Elastollan > Styroflex lowest) were dependent on the surface characteristics, hydrophilicity, and surface charge of the material. These surface parameters might be predictive for minimally encrusting surfaces and allow for quick screening of novel materials.

We suggest considering encrustation experiments for the development of new ureteral stents to minimize encrustation.

## Figures and Tables

**Figure 1 polymers-12-00558-f001:**
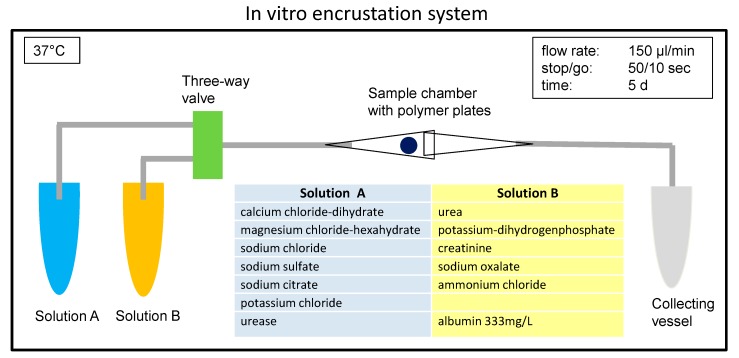
Scheme of the in vitro encrustation system. All components are placed in a heating chamber to maintain 37 °C. Solutions A+B are mixed in the three-way valve, subsequently leading to encrustations crystallizing on the polymer pieces placed in the sample compartment. Formulation of the synthetic urine adopted from [8].

**Figure 2 polymers-12-00558-f002:**
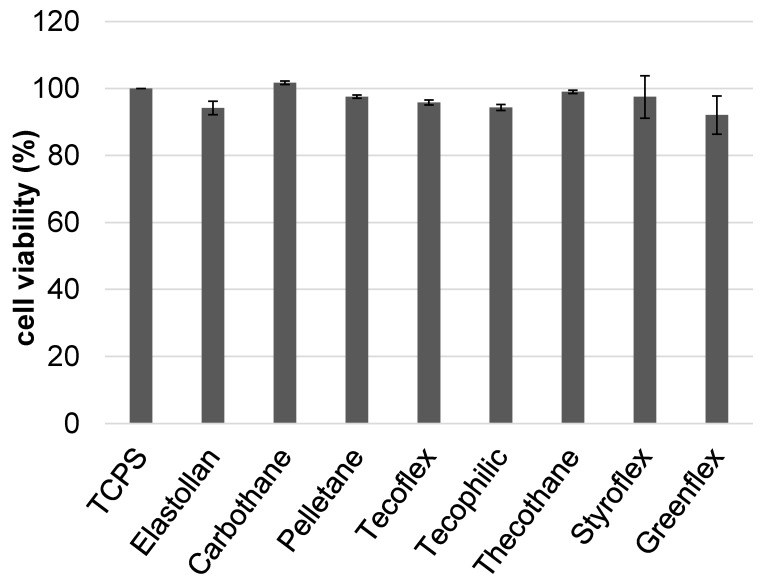
Biocompatibility of the tested polymer samples. Urothelial cells are not impaired by any of the polymers tested (100% reference = tissue culture polystyrene (TCPS), HUC-1 cells, mean ± SD, ANOVA post hoc Bonferroni n.s. at *p* < 0.05).

**Figure 3 polymers-12-00558-f003:**
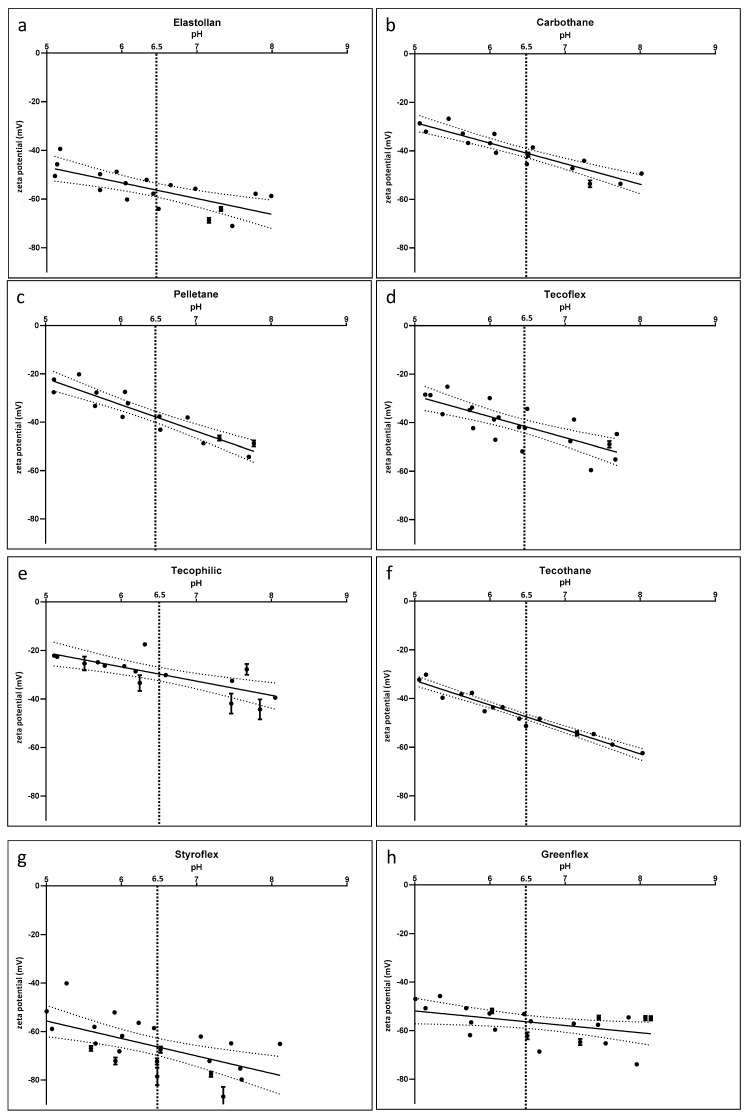
Zeta potential of the polymers Elastollan (**a**), Carbothane (**b**), Pelletane (**c**), Tecoflex (**d**), Thecophilic (**e**), Tecothane (**f**), Styroflex (**g**) and Greenflex (**h**). The absolute values show a variation (with lowest absolute values for Styroflex, Greenflex, and Elastollan), while the slope of the curves is similar for all samples: decreasing zeta potential with increasing pH. pH 6.5 marks the average pH for urine. mean ± SD. Solid line marks mean, while dotted line shows the 75% confidence interval. n = 3 independent samples (SurPASS, Anton Paar).

**Figure 4 polymers-12-00558-f004:**
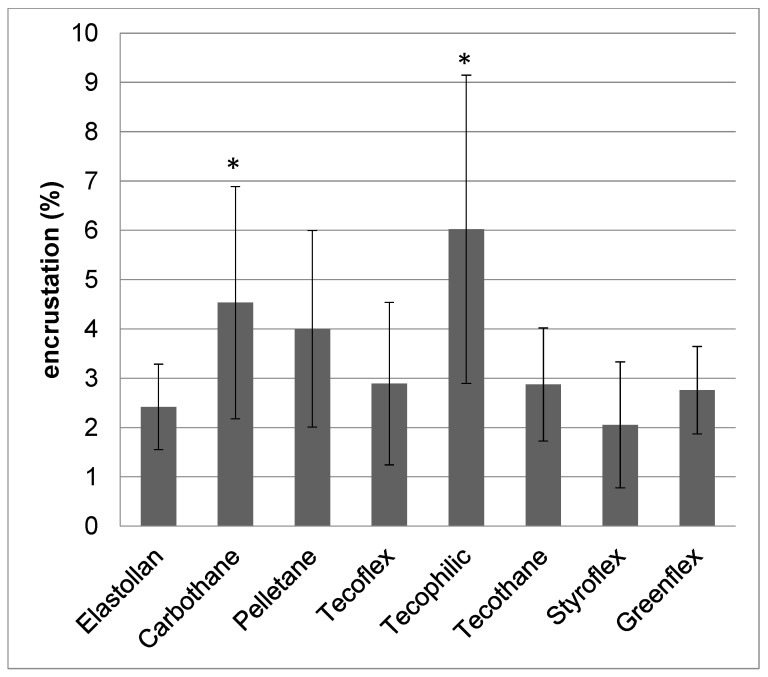
Quantification of encrustations on polymer bulk materials after 5 d in the in vitro encrustation system. n ≥ 3. mean ± s.e.m., statistics: Kruskal–Wallis post hoc, uncorrected Dunn’s test, * *p* < 0.05 compared to Elastollan.

**Figure 5 polymers-12-00558-f005:**
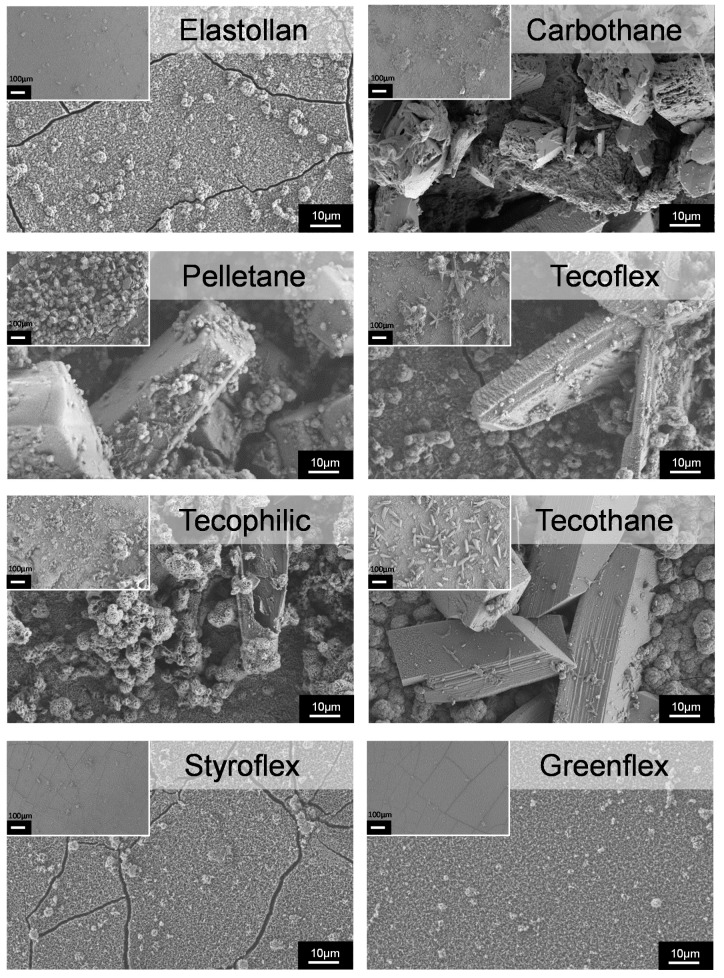
SEM images of encrusted polymer surfaces after 5 d in the in vitro encrustation system. Note the mix of amorphous depositions and clear-cut crystals. The surface of Elastollan, Styroflex, and Greenflex is covered by only a thin layer of amorphous depositions, whereby nearly no large crystals are present. The other polymers (Carbothane, Pelletane, Tecoflex, Thecophilic, Tecothane) show large crystalline depositions on the surface (bar 10 µm, bar insert 100 µm; FESEM Merlin, Zeiss).

**Figure 6 polymers-12-00558-f006:**
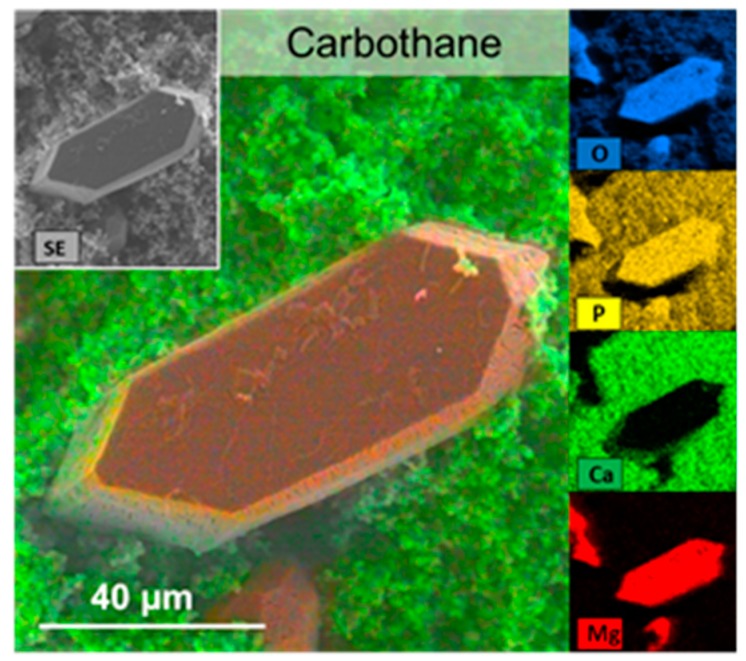
EDX analyses of encrustations on Carbothane. Mapping of the elements shows large crystals of Mg, P, and O, presumably being struvite (appearing orange). The Ca- and P-containing matrix of calcium carbonate apatite (appearing green) was multi-layered and rough in nature. FE-SEM Merlin VP compact (Zeiss) with EDX-detector XFlash 6/30 (Bruker).

**Figure 7 polymers-12-00558-f007:**
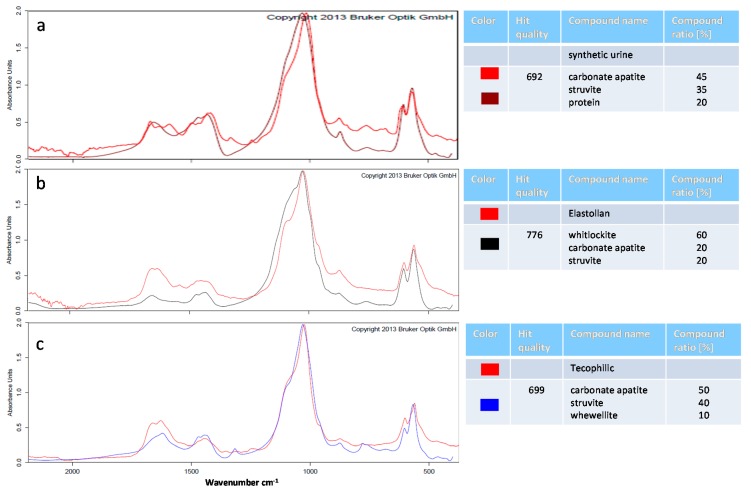
FTIR analysis of encrustations formed in the in vitro-encrustation system. Measured absorption spectra (red) of mineral deposits of the synthetic urine without polymer (**a**) or the precipitations on the polymer samples (Elastollan: **b**, Tecophilic: **c**). Absorption spectra of the OPUS™ reference library with the best match are shown in purple, black, and blue, respectively. The amount of calcium phosphate (carbonate apatite and whitlockite) precipitates is higher on Elastollan. In contrast, on Tecophilic, the amount of magnesium phosphates (struvite) is elevated compared to complete urine precipitates. (ALPHA FTIR spectrometer, OPUS^TM^ library, Bruker).

**Table 1 polymers-12-00558-t001:** List of the polymer types and processing parameters to produce flat samples.

Polymer Type	Trade Name	Processing Temperature (°C)	Drying Temperature (°C)	Drying Process (h)
Thermoplastic Urethanes	Elastollan 1185A 10 FC	188	100–111	2–4
	Carbothane PC 3585A	193	65 (57)	4-overnight
	Pelletane 2363 80A	198	80–95	2–4
	Tecoflex EG 60D	193	65.5 (54.5)	2-overnight
	Tecophilic HP 93A 100	182	65.5 (54.5)	2-overnight
	Tecothane TT 1095	205	95 (80)	2-overnight
Styrene-Butadiene Copolymer	Styroflex 2G 66	235	90	2–4
Ethylen-Vinylacetat	Greenflex FL 65	103	70	3

**Table 2 polymers-12-00558-t002:** Material characteristics of different polymer types. The quantity of encrustation can be correlated with the negative zeta potential and slight hydrophilicity of the polymer. Fields marked green indicate the lowest values for each parameter. Mean ± SD.

Material	Contact Angle (°) at pH 6.5		Surface Charge (mV) at pH 6.5		Encrustation (%)	
Elastollan	85	±13	−56.5	±2.8	2.4	±1.2
Carbothane	92	±2	−41.2	±1.9	4.5	±3.3
Pelletane	105	±5	−38.2	±2.3	4	±2.8
Tecoflex	111	±2	−41.8	±2.8	2.9	±2.3
Tecophilic	104	±6	−29.6	±2.8	6.0	±4.4
Tecothane	104	±4	−47.9	±1.2	2.9	±1.6
Styroflex	85	±5	−66.6	±3.6	2.1	±1.8
Greenflex	86	±1	−56.4	±2.7	2.8	±1.2
	n = 3–10 independent replicates		n ≥ 3 independent replicates		n = 3 independent replicates	

## Data Availability

The raw data required to reproduce these findings are available to download from Mendeley Data.

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
