# Peer review of "Prevention of Encrustation on Ureteral Stents: Which Surface Parameters Provide Guidance for the Development of Novel Stent Materials?"

_polymers, 2020, doi:10.3390/polym12030558_

Round 1

Reviewer 1 Report

It seems that the work is hard to give a quantitative guidance. For example, in the Conclusion, the authors mentioned that "materials with a moderate wettability". What exactly does "moderate" mean for future references? It is not surprising that multiple parameters affect the encrustation, does the author have a conclusive summary on how to determine this with values, meanwhile integrating different parameters?

Reviewer 2 Report

This is a very interesting study demonstrating the role of material surface parameters in guiding encrustation in ureteral stents. The authors characterized an array of biocompatible polymers for its surface charge and wettability and correlated it to the material's tendency to encrust. More interestingly, they utilized synthetic urine supplemented with bacterial enzyme to mimic the infection in a near physiological manner. Overall, the manuscript is well written, and the experiments are designed and executed well.  This study has direct implications for the development of novel ureteral stents with improved surface properties to minimize encrustation. Hence, I recommend accepting the manuscript after some minor corrections that are provided below;

  • It would be good if the authors can discuss more about the proposed mechanism by which strong negative surface charge reduces encrustation.
  • In Figure 2, please change vitality to viability
  • Please remove +/- SD or SEM from the X-axis label and add it to the figure legend.
  • Page 9, Line 254 Mean +/- SD
  • In Figure 7, please label all the characteristic peaks in the spectra relevant to the compounds analyzed.
  • Please explain in the methodology about the specific FTIR peaks considered for the quantification of the percentage of various calcium phosphate phases.
